# Anisotropy of Electrical and Thermal Conductivity in High-Density Graphite Foils

**DOI:** 10.3390/nano14131162

**Published:** 2024-07-07

**Authors:** Vladimir A. Shulyak, Nikolai S. Morozov, Alexandra V. Gracheva, Maria D. Gritskevich, Sergei N. Chebotarev, Viktor V. Avdeev

**Affiliations:** Department of Chemistry, Lomonosov Moscow State University, Moscow 119991, Russia; shulyak.v@unichimtek.ru (V.A.S.); gracheva.a@inumit.ru (A.V.G.); mdgritskevich@mephi.ru (M.D.G.); chebotarev.s@inumit.ru (S.N.C.); avdeev@highp.chem.msu.ru (V.V.A.)

**Keywords:** flexible graphite foil, electrical conductivity, thermal conductivity, anisotropy, structural parameters, Kearns parameters

## Abstract

Flexible graphite foils with varying thicknesses (S = 282 ± 5 μm, M = 494 ± 7 μm, L = 746 ± 8 μm) and an initial density of 0.70 g/cm^3^ were obtained using the nitrate method. The specific electrical and thermal conductivity of these foils were investigated. As the density increased from 0.70 g/cm^3^ to 1.75 g/cm^3^, the specific electrical conductivity increased from 69 to 192 kS/m and the thermal conductivity increased from 109 to 326 W/(m·K) due to the rolling of graphite foils. The study showed that conductivity and anisotropy depend on the shape, orientation, and contact area of thermally expanded graphite (TEG) mesoparticles (mesostructural factor), and the crystal structure of nanocrystallites (nanostructural factor). A proposed mesostructural model explained these increases, with denser foils showing elongated, narrowed TEG particles and larger contact areas, confirmed by electron microscopy results. For graphite foils 200 and 750 μm thick, increased density led to a larger coherent scattering region, likely due to the rotation of graphite mesoparticles under mechanical action, while thinner foils (<200 μm) with densities > 1.7 g/cm^3^ showed increased plastic deformation, indicated by a sharp reduction in the coherent scattering region size. This was also evident from the decrease in misorientation angles with increasing density. Rolling reduced nanocrystallite misorientation angles along the rolling direction compared to the transverse direction (TD) (for 1.75 g/cm^3^ density ΔMA = 1.2° (*S*), 2.6° (*M*), and 2.4° (*L*)), explaining the observed anisotropy in the electrical and mechanical properties of the rolled graphite foils. X-ray analysis confirmed the preferred nanocrystallite orientation and anisotropy coefficients (A) using Kearns parameters, which aligned well with experimental measurements (for L series foils calculated as: A_0.70_ = 1.05, A_1.30_ = 1.10, and A_1.75_ = 1.16). These calculated values corresponded well with the experimental measurements of specific electrical conductivity, where the anisotropy coefficient changed from 1.00 to 1.16 and mechanical properties varied from 0.98 to 1.13.

## 1. Introduction

Flexible graphite foils (GFs) are a derivative of natural graphite that retains high chemical and thermal resistance [1,2] and conducts electric current [3,4,5,6]. Additionally, GFs acquire flexibility and mechanical strength, expanding the already considerable range of industrial applications for graphite materials.

Dense foils (>1.5 g/cm^3^) exhibit high electrical and thermal conductivity, enabling their use in Li-ion batteries for heat dissipation and current transmission with appropriate processing [7,8], which reduces the weight of battery designs. High-density graphite foils can also be used to investigate magnetic properties under specific conditions at high temperatures for applications in quantum technologies, similar to graphene [9,10]. Low-density graphite foils (<0.7 g/cm^3^) have low thermal conductivity [11], making them suitable as passive temperature regulators, significantly reducing harmful emissions by eliminating the need for additional heating or cooling in buildings.

GFs are produced through chemical–physical action on flake graphite. Direct treatment of crystalline graphite with strong Brønsted acids and, if necessary, oxidants leads to the formation of a graphite intercalation compound (GIC) [11,12,13]. Subsequent hydrolysis and heating of the resulting material contribute to the shock removal of the embedded substances from the matrix [14,15]. By adjusting the rate and magnitude of heating, it is possible to achieve conditions under which the graphite powder transforms into thermally expanded graphite (TEG), a material with a significantly different morphology [15,16,17,18,19,20,21,22]. The same transformation can also be achieved by other methods [23,24]. The TEG feature is its ability to form mechanically strong, dense compacts (GFs) while retaining the crystal lattice of natural graphite.

Carbon materials are headliners in thermal conductivity. The record holder in this field is graphene, capable of developing λ ≈ 4800–5300 W/(m·K) [25,26,27]. Examples of other materials with high values are: diamonds Type IIa—λ ≈ 2300 [28,29] and Type IIb—λ ≈ 1350 W/(m·K), which are significantly higher relative to classical conductors such as copper λ ≈ 400 W/(m·K) [30,31] and aluminum λ ≈ 220 W/(m·K) [24,25]. The starting point for this study was the pyrolytic quasi-monocrystal of graphite with the thermal conductivity 2000–2300 W/(m·K) [32,33,34,35]. Unlike the aforementioned carbon materials, natural graphite is more defective but has the greatest industrial potential due to its relatively low cost, ease of manufacturing, and high production volume.

The rolling process induces various structural changes at both the mesoscopic and nanoscopic levels within graphite foils. These structural modifications play a crucial role in determining the thermal and electrical conductivities of the foils. However, to date the data on these alterations have not been studied enough, making it challenging to create accurate computational models that can predict the final properties of the GF products.

Generally, the research on the anisotropy of graphite foils is conducted parallel to the rolling plane and along the normal to it. However, there is a significant gap in the understanding of anisotropy within the rolling plane itself, particularly when comparing properties along the rolling direction to those in the transverse direction. To address this gap, it is essential to study the crystallographic texture of rolled graphite foils. GFs consist of nanocrystallites with a hexagonal crystal lattice, making them particularly suitable for this type of analysis. The crystallographic texture can be characterized by determining the Kearns parameters, which quantify the properties values and their anisotropy. Despite the relevance of Kearns parameters in understanding the anisotropy of graphite materials, this area has received little attention in existing research. Thus, a thorough investigation of the crystallographic texture using Kearns factors is crucial for the comprehensive understanding of GFs’ anisotropic properties.

This paper examined the thermal and electrical conductivity of GFs of various thicknesses and densities. The influence of mesostructural and nanostructural characteristics on the material physical and mechanical properties was determined. A simplified model was presented to account for these characteristics. The texture method was used to determine the anisotropy of these properties according to the Kearns parameters.

Thus, the purpose of this work is to determine the influence of mesostructural and nanostructural factors on the physical and mechanical properties of GFs and their anisotropy in the rolling plane.

## 2. Sample Preparation and Test Methods

### 2.1. Production of Flexible Graphite Foils

GFs were produced by rolling TEG, which was formed by the thermal decomposition of exhaust gas from graphite nitrate [36]. The primary method for producing TEG is through the intercalation of graphite with sulfuric acid [37,38,39,40]. However, in this study, graphite was intercalated with nitric acid, utilizing the nitrate method to produce a graphite intercalation compound GIC [41].

The process of obtaining graphite samples included several steps. Initially, natural graphite was treated with fuming nitric acid (it was kept for 30 min) in a weight ratio of 1:0.8 to obtain the GIC. The GIC was then hydrolyzed and dried, resulting in oxidized graphite adduct. The next step was foaming the exhaust gas at a temperature of 900 °C to produce TEG. The resulting foamed graphite was rolled to obtain GFs produced by the nitrate method (Figure 1a). A more detailed description of the production process of the studied material is presented in our earlier publication [42].

Three GF series were prepared to study specific electrical and thermal conductivity. Initially, three GF samples were obtained from TEG powder with a density of 0.70 g/cm^3^ and thicknesses of 282 μm (small—*S* series), 494 μm (medium—*M* series), and 746 μm (large—*L* series), which were then additionally rolled to densities of ~1.00, 1.30, 1.60, and 1.75 g/cm^3^ resulting in a corresponding decrease in their thickness. In total, 15 GF samples were prepared: 3 series with 5 samples of each.

The accurate dimensions of the GFs are given in Table 1. The thickness changes are illustrated in Figure 1b, which shows SEM images with the indicated thicknesses of S, M, and L series foil samples with densities of 0.70, 1.30, and 1.75 g/cm^3^. In the text, the samples are designated by their series and density value (for example, S-0.70).

Samples in the form of strips cut along the RD (Figure 1a, blue) and across the RD (Figure 1a, turquoise) were examined to obtain information on their electrically conductive properties. Round samples were examined to obtain information on their heat-conducting properties (Figure 1a, red).

### 2.2. Bulk Density Determination of GFs

The test samples’ densities were determined by the simple ratio (1) for different volume samples (Equations (2) and (3)):(1)$$d=\frac{m}{V};$$
(2)$$V_{term}=\frac{\pi \cdot D^{2}}{4};$$
(3)$$V_{res}=l\cdot b\cdot h;$$
where d is the GF’s density; m is the mass of the measured sample; $V_{term}$ is the volume of the sample for the study of heat-conducting properties; D is the diameter of the sample; $V_{res}$ is the volume of the sample for the study of electrically conductive properties; and l, b, h is the length, width, and thickness of the samples for the study of electrical conductivity, respectively.

For round samples, D was measured in two perpendicular directions (measurement error ≤ 10 μm), and the obtained values were averaged. The thickness was measured (measurement error ≤ 4 μm) five times—once in the center and once on four opposite sides. The obtained values were also averaged. For strip samples, the length and width values were averaged over three measurements, while thickness was measured similarly to round samples. The weight of each sample from the collection was determined several times on a Shimadzu AX200 scale (Kyoto, Japan). The measurement error was ≤0.001 g.

This approach enabled an accuracy of 1% in determining the density for the *S* series. In this series, deviations in geometric dimensions and mass significantly impact the measurement error compared to the *M* and *L* series. Therefore, the density measurement error for all samples in this study was assumed to be 1%.

### 2.3. Scanning and Transmission Electron Microscopy Examination of Foil Samples

The microstructure of the GFs was examined by scanning electron microscopy (SEM) using TESCAN VEGA 3 (Brno, Czech Republic). The electron gun of the device used a cathode made of lanthanum hexaboride. The electron beam was formed at the accelerating voltage HV = 20 kV. The vacuum in the chamber was 4 × 10^−4^ Pa. The secondary electron detector SE was used. SEM images were obtained with the view field 1000 × 1000 μm. Image processing was performed using the software provided with the equipment.

The image of the mesostructure (the graphite worm-like flakes and the values of the angles between them) and nanostructure (the nanocrystallites’ size) was examined by transmission electron microscopy using a TITAN™ TEM (Hillsboro, OR, USA). Lamellas were prepared using SEM FEI SCIOS 2 (Hillsboro, OR, USA).

### 2.4. XRD Analysis of the Test Samples

#### 2.4.1. Structural Studies of GF Nanocrystallites

The study of the structural characteristics of the GF material was carried out by X-ray diffraction (XRD) analysis using a Rigaku Ultima IV diffractometer (Tokyo, Japan). The geometry of the device corresponded to Bragg–Brentano focusing. X-ray tube radiation CuK_α_ = 0.15418 nm was used. The XRD patterns of the GF samples of the *S*, *M*, and *L* series with densities of 0.70, 1.30, and 1.75 g/cm^3^ were recorded in the scanning mode of the *θ*/2*θ* of the diffractometer in Bragg angles 2*θ* = 20–90°, in increments of 0.02°, and with a scanning speed of 2°/min. The XRD pattern of the *LaB*_6_ standard sample was recorded under similar conditions to account for the instrumental contribution to the measurement error. Data analysis was conducted using PDXL2 (ver. 2.8.40) software, which was supplied with the equipment.

#### 2.4.2. Coherent Scattering Region Sizes and Microstresses in GF Nanocrystallites

To assess the structural changes in graphite, the characteristics of reflexes in XRD patterns were analyzed: their integral intensity ($I_{int}$), the full width at half-maximum of its height (*FWHM*), and the reflex positions, which reflect the value of the interplanar distance $d_{cryst}$. In crystalline specimens undergoing mechano-thermal treatment, the *FWHM* is a convolution of contributions from different constituents. In the simplified version, this is described by Equation (4). This material was subjected to external deformations resulting from cold rolling; therefore, the calculation of the coherent scattering regions (CSRs) by the Scherrer method was not applicable. Instead, the Williamson–Hall method was used, which accounts for the effect of microdeformations in determining the CSRs ($L_{c}$) [43,44,45], which characterize the GF nanocrystallites’ size (Equation (6)):(4)$$FWHM=\beta_{L}+\beta_{\varepsilon }+\beta_{instr}$$
(5)$$\beta_{phys}=FWHM-\beta_{instr}=\beta_{L}+\beta_{\varepsilon }=\frac{0.9\cdot \lambda }{L_{c}\cdot \cos\theta }+4\cdot \varepsilon \cdot \tan\theta ;$$
(6)$$\beta_{phys}\cdot \cos\theta =\frac{0.9\cdot \lambda }{L_{c}}+4\cdot \varepsilon \cdot \sin\theta ;$$
where FWHM is the value of the full width at half-maximum of the reflex of the experimental XRD pattern; $\beta_{phys}$ is the physical broadening of the reflex caused by the material structural features; $\beta_{instr}$ is the contribution of the instrumental line broadening to FWHM caused by the diffractometer geometry and characteristics; $\beta_{L}$ is the contribution of the broadening of the X-ray reflection caused by the fine dispersion of the sample nanocrystallites; $\beta_{\varepsilon }$ is the contribution of the reflex broadening arising from the presence of microdeformations; λ is the wavelength of X-ray radiation; $L_{c}$ is the size of the CSRs along the parameter from the crystal lattice of graphite; θ is the angle of incidence of X-ray radiation; and ε is microdeformations in nanocrystallites caused by machining processes.

The CSR calculations were carried out only along the parameter *c* of the crystal lattice by reflections (00*.l*). The ${(\beta }_{phys}\cdot \cos\left(\theta \right))$ dependence on (4·sin⁡θ) was built, which was approximated by a linear function and extrapolated to intersect with the ordinate axis. The microstresses in the nanocrystals were determined by the tangent of the inclination angle and the size of the CSR was determined by the value at the intersection point of the ordinate axis with the trend line.

#### 2.4.3. Determination of the Misorientation Angles of Nanocrystallites by the Rocking Curves

The misorientation angles (*MA*) of nanocrystallites were determined by X-ray [46]. For this, rocking curves were recorded to (00.6) to reflect XRD patterns of hexagonal graphite on the Rigaku Ultima IV diffractometer. X-ray tube radiation was *CuK_α_* = 0.15426 nm and Bragg–Brentano focusing was used. The recording was carried out in the diffractometer *θ* mode at a fixed value of 2*θ* (00.6) reflection XRD patterns for GFs of the *S*, *M*, and *L* series with densities of 0.70, 1.30, and 1.75 when scanning at *θ* from 10° to 80°. Then, the *FWHM* of the resulting rocking curves was calculated using PDXL2 software, which was taken as the MA value.

#### 2.4.4. The Direct Pole Figure and The Orientation Distribution Function Analysis of the Test Samples

GFs are subjected to cold rolling, which can result in a preferred crystallographic orientation of nanocrystallites. The X-ray method of recording incomplete direct pole figures (DPFs) was used to qualitatively and quantitatively describe this orientation [47]. The orientations of the specimen are determined by its successive inclination by an angle ψ and rotation around the normal line to the surface by an angle φ. The geometric features of such a survey limit the angle change ψ within 0–80°, while the rotation angle φ can vary within 0–360°, as a result of which the stereographic projection is not completely filled, and incomplete DPFs were obtained (Figure 2a). Also, when constructing the DPFs, the values of the background and the defocusing coefficient were determined. The analytical Equation (7) for constructing the DPFs was followed:(7)$$P_{ij}=[I_{ij}-I_{iBG}]\cdot K_{i}/\bar{I},$$
(8)$$\bar{I}=\int_{0}^{2\pi }\int_{0}^{\psi_{max}}K_{\psi }\cdot [I(\psi ,\varphi )-I_{\psi }]\cdot \sin\psi \cdot d\psi \cdot d\varphi /\int_{0}^{2\pi }\int_{0}^{\psi_{max}}\sin\psi \cdot d\psi \cdot d\varphi ;$$
where $P_{ij}$ is the value of the pole density on the stereographic projection, $I_{ij}$ is the value of the measured intensity at successive points of the orientation space ($\psi_{i}$,$\;\varphi_{j}$); $I_{iBG}$ and $I_{\psi }$ are the values of the sample background intensity for the corresponding inclination angle $\psi_{i}$; and $K_{\psi }$ is the value of the defocusing correction coefficient, also varying with the inclination angle $\psi_{i}$.

The complete texture description of the material can be obtained through the calculation of the orientation distribution function (ODF) of nanocrystallites (Equation (9)) [48] using MTEX (ver. 5.11.2) free software. This model contains comprehensive information about all orientations of nanocrystallites within the sample volume under study. The ODF allows for the calculation of any property of the material from the properties of individual crystallites. The ODF (*f*(*g*)) shows how the crystallites in the polycrystalline are distributed over the crystallographic orientations *g* described by Euler angles (*φ*_1_, Φ, *φ*_2_) (Figure 2c). These angles reflect the relation between the $K_{a}$ coordinate system related to the study sample and the $K_{b}$ system, which is related to the crystallographic orientation of the grains in the material (Figure 2b).

The ODF is the four-dimensional function: three coordinates (Euler angles) determine the crystallographic orientation of the grains, and the fourth is the distribution density (*P*). In this case, the distribution density is understood as the probability of finding grains in the sample volume under study that have an orientation *g*. It is calculated using incomplete DPFs, which are a three-dimensional projection of the ODF. The connection between the DPFs and the ODF was carried out through integral Equation (10):(9)$$\int f\left(g\right)dg=\frac{1}{8\pi^{2}}\int_{0}^{2\pi }\int_{0}^{\pi }\int_{0}^{2\pi }f\left(\varphi_{1},\Phi ,\varphi_{2}\right)d\varphi_{1}\sin\Phi d\Phi d\varphi_{2}=1$$
(10)$$SPF:P_{hkl}(\psi ,\varphi )=\frac{1}{2\pi }\int_{\left(\psi ,\varphi \right)}^{max}f(g)d\psi$$
where $P_{hkl}$ is the pole density from the normal line to the (*hkl*) plane; f(g) is the ODF; $\varphi_{1},\Phi ,\varphi_{2}$ are Euler angles; and *g* is the crystallographic orientation described by the corresponding Euler angles.

#### 2.4.5. Kearns Parameters Calculation for GFs

The X-ray method was applied to calculate the anisotropy of the GFs’ properties. It was calculated with the averaged values of the single crystal (or pyrolytic) graphite properties and nanocrystal volume fractions according to the experimental integral texture Kearns parameters (f-parameters) for GFs [49,50]. Kearns parameters are the sum projections of the basis [00.1] normal volume fractions on the selected direction in the sample. The value of some polycrystalline material property PP in the direction ψ was determined by averaging the nanocrystals’ (single crystals) properties in this direction according to their volume fraction $V_{i}$ (Equation (11)) [51,52]. If one neglects the interaction of grains in the polycrystal, then this property value is the properties’ sum of the *i*-th nanocrystallites [53]:(11)$${PP}_{\psi }={PP}_{c}\cdot \sum_{i}V_{i}{cos}^{2}\psi_{i}+{PP}_{a}\cdot \sum_{i}V_{i}(1-{cos}^{2}\psi_{i})=f\cdot {PP}_{c}+(1-f)\cdot {PP}_{a},$$
where ${PP}_{c}$ and ${PP}_{a}$ are the properties along the axes of the crystal lattice c and a, respectively; and $\psi_{i}$ is the angle of deviation of the selected direction of the *i*-th grain from the normal line to the reference plane, i.e., from the axis *c*.

Thus, knowing the properties of pyrolytic graphite and the f-parameter values, it can be possible to calculate the anisotropy of various physical and mechanical properties of the sample (for example thermal and electrical conductivity) by Equation (11).

### 2.5. Electrical Characteristics Determination

The modified four-probe van der Pauw method was used for measuring the specific electrical conductivity. GPD-73303D DC power source and GDM-78255A voltmeter (both by GW Instek, Taipei, Taiwan) were used for obtaining current–voltage characteristics (Figure 3).

The GF elementary sample with a length of 50 ± 1 mm and a width of 10 ± 0.5 mm was located on two pairs of platinum contacts at a temperature of 25 °C and was pressed with a slight force to fix the sample and ensure stable electrical contact. The first pair of contacts (Figure 3, red) supplied direct current to the sample, while the second pair (Figure 3, blue) measured the voltage. By cutting samples in the (RD) and (TD) as indicated in Figure 1, it was possible to determine the characteristics of the sample depending on the measurement direction and calculate the anisotropy coefficient of the material.

The measurement process began with determining the sample resistance by constructing the current–voltage characteristic in the current strength range from 0.1 to 0.5 A. To compensate for the Seebeck effect, the mean voltage value from measurements in two opposite directions of current flow was used. The sample resistivity was determined by Equation (12):(12)$$\rho =\frac{R\cdot (b\cdot h)}{l}$$
where ρ is electrical resistivity (Ohm·m); R is the resistance of the tested sample (Ohm); b is the width of the sample (m); h is the thickness of the sample (m); and l is the length between the contacts of the measured section (const, l = 40 mm).

Electrical conductivity is determined by Equation (13):(13)σ=1/ρ
where σ is electrical conductivity (S/m); ρ is electrical resistivity (Oh·m).

Depending on the installation of the test sample, the error was defined for current as: 2.27–4.50%, and for measured voltage as: 0.03–0.09%. The resulting error in determining resistivity was calculated through the maximum deviations of current and voltage and amounted to 4.5%.

### 2.6. Thermal Diffusivity and Thermal Conductivity Determination

Thermal diffusivity was measured by the light flash method using NETZSCH LFA467 (Bremen, Germany) at 25 °C in the In-plane direction. This method involves measuring the temperature response on one surface of the sample from the moment it was heated by the light flash on the opposite side. The results were processed using the NETZSCH Proteus software (https://analyzing-testing.netzsch.com/en/products/software/proteus, accessed on 4 July 2024).

The In-plane method fixes the temperature response at a certain distance from the center of the sample while blocking the rest of the radiation with a mask. This approach allows for determining the thermal diffusivity in the plane. The mask is a disk with four windows spaced apart from the center and located at 90° relative to each other (Figure 4). Consequently, the measured value is the average over the entire plane, and this method does not establish the anisotropy of the material in the plane.

The measurement procedure is presented in Figure 4b. Initially, the background radiation from the sample was recorded and taken as zero energy. At point (1) from Figure 4b, the light flash was fired, starting the time measurement. After a while, the energy reaches its maximum (E_max_) at point (2) from Figure 4b. To determine the final parameter, it is necessary to determine the time (τ_1/2_), which corresponds to half the maximum energy (E_max_/2) at point (3) from Figure 4b. The calculation of thermal diffusivity was carried out automatically using software according to the laws established in the paper [54].

The calculation of thermal conductivity was carried out according to the following Equation (14):(14)λ=a·c·d
where λ is thermal conductivity, W/(m·K); a is thermal diffusivity, mm^2^/s; *c* is specific heat at 25 °C, J/(g·K), for which the value of 0.711 J/(g·K) from the literature [55,56] was used; and d is the bulk density of the test sample, g/cm^3^.

The error in measuring thermal diffusivity was less than 3%, and thermal conductivity was less than 5%.

## 3. Results and Discussions

### 3.1. Study of Thermal and Electrical Properties in the GFs

#### 3.1.1. Electrical Characteristics of GFs

To study the specific electrical conductivity of the GFs, samples from each series (*S*, *M*, *L*) with densities of 0.70, 1.00, 1.30, 1.60, and 1.75 g/cm^3^ were installed in the holder of the installation described in clause 2.6. The resistance *R* of these samples was measured by recording the current–voltage characteristic. Specific electrical resistivity ρ was calculated by using Equation (12). The specific electrical resistance of GF samples of all *S*, *M*, and *L* series with densities from 0.70 g/cm^3^ to 1.75 g/cm^3^ decreased significantly with increasing density, both in the RD (Figure 5, green bars) and in the TD (Figure 5, light green bars).

These dependencies were described by the power function, the equations of which are presented in (15) and (16) with their determination coefficients (dotted green lines).
(15)$$\rho_{RD}\left(d\right)=9.830\cdot d^{-1.082},\;R^{2}=0.9999;$$
(16)$$\rho_{TD}\left(d\right)=9.830\cdot d^{-1.041},\;R^{2}=0.9961.$$

The nature of these degree dependencies was due to the fact that the studied specific materials’ characteristics were related to their physical dimensions, which cannot be ≤0. The data analysis indicated that the foils’ electrical resistivity values were decreased by more than 2.5 times as density increased from 0.70 g/cm^3^ to 1.75 g/cm^3^: the values were decreased from 14.6 to 5.6 Ohm·m.

Specific electrical conductivity *σ* is the inverse dependence on specific resistance (Equation (13)). Consequently, as the density increased from 0.70 to 1.75 g/cm^3^ in the GFs of all *S*, *M*, and *L* series, there was an increase in σ from 69 kS/m to 187 kS/m in the RD (Figure 5, blue bars) and from 67 kS/m to 169 kS/m in the TD (Figure 5, light blue bars). This parameter is described by the linear dependence. The equations and determination coefficients for *σ* are presented in (17) and (18) (dotted blue lines):(17)$$\sigma_{RD}\left(d\right)=111.7\cdot d-9.461,\;R^{2}=0.9997;$$
(18)$$\sigma_{TD}\left(d\right)=99.60\cdot d-0.896,\;R^{2}=0.9931.$$

In the density range of (0.70–1.75 g/cm^3^), there was no clearly expressed power dependence for specific electrical conductivity. This is because the nature of the power dependence for specific electrical conductivity becomes apparent at higher values of GF density [57].

For both electrical resistivity and conductivity at high densities, there was a notable difference in values between the RD and TD, attributable to the anisotropy of the properties.

#### 3.1.2. Thermal Characteristics of GFs

When studying the thermal characteristics of GF samples of the *S*, *M*, and *L* series with densities from 0.7 g/cm^3^ to 1.75 g/cm^3^ using the LFA 437 and the In-plane method, the dependencies of thermal diffusivity on their density were obtained, as shown in Figure 6.

The thermal diffusivity of the GFs was increased: from 217 mm^2^/s at a density of 0.70 g/cm^3^ to 266 mm^2^/s at a density of 1.75 g/cm^3^. At lower densities, there was a noticeable difference in the values of the parameter depending on the thickness of the GFs: at d = 0.70 g/cm^3^ for 11 mm^2^/s. With an increase in density, the difference in thermal diffusivity decreased (at d = 1.75 g/cm^3^ for 5 mm^2^/s). This effect can be attributed to the homogenization of pores as density increases. The temperature diffusivity change from density varied linearly (dotted orange line), and is described by Equation (19):(19)$$a\left(d\right)=47.3\cdot d-184.1,\;R^{2}=0.9983.$$

A more explicit increase in thermal characteristics was observed by the thermal conductivity coefficient, which also depended on the GF density for all series. On average, thermal conductivity was increased three times with the density rise: from 109 W/(m·K) at density 0.70 g/cm^3^ to 326 W/(m·K) at density 1.75 g/cm^3^, and is described by Equation (20) (dotted red line):(20)$$\lambda \left(d\right)=212.7\cdot d-45.4,\;R^{2}=0.9991.$$

This parameter was more suitable for assessing the thermal properties of the material, since the thermal conductivity was dependent not only on the structural characteristics of the graphite crystal lattice, but also on its mesoparameters (the density and the specific heat). This was evident from the minimal differences of thermal conductivity coefficients among the *S*, *M*, and *L* series at the fixed density.

#### 3.1.3. The Interrelation between Thermal and Specific Electrical Conductivity

The interrelation between thermal and specific electrical conductivity was determined. It was expressed in terms of the proportionality constant (Lorentz number) from the Wiedemann–Franz law (21):(21)$$\frac{\lambda }{\sigma }=LT,$$
where λ is the thermal conductivity of graphite, W/(m·K); σ is the specific electrical conductivity, S/m; L is the Lorentz number, Ohm·W/K^2^; and T is the absolute temperature, K.

The In-plane laser flash method was used to determine the average thermal conductivity in the GF rolling plane, then the electrical conductivity was averaged. The results are shown in Figure 7. The resulting error for the Lorentz number was 6.7%.

The Lorenz number directly characterizes the material’s state. This parameter depends on the structural parameters of the crystallites and should not vary with the material’s porosity. However, data analysis revealed a slight trend of an increasing proportionality constant with increasing density of graphite foils. This suggests that nanocrystallites undergo structural changes that affect this parameter.

#### 3.1.4. Peculiarities of the Thermal and Electrical Properties of the GFs

The change in the specific electrical and thermal conductivity slightly was dependent on the GFs’ thickness (*S*, *M*, and *L* series at the densities from 0.70 to 1.75 g/cm^3^). Apparently, the nature of their distribution was also influenced by internal changes during rolling. This means that it was dependent on meso-(density/porosity) and microstructural (crystallographic orientation and size of nanocrystallites) transformations, which impacted the GFs’ final properties.

The observed phenomena of heat and electric charge transfer with increasing GF density showed a consistent trend for the *S*, *M*, and *L* series: both the specific electrical and thermal conductivity increased. The unit cross-sectional area of the conductor sample became more was saturated with matter (increased density), which increased its specific electrical conductivity. A similar effect was seen in thermal conductivity. Newly formed contacts between individual particles reduced thermal resistance, thereby increasing the maximum density of the heat flux.

The more complex mechanism in thermal diffusivity was observed: at the minimum density 0.70 g/cm^3^ on a 217 mm/s^2^ and at density 1.75 g/cm^3^ on a 266 mm/s^2^ (an increase of only 22%). This increase was associated with structural changes in the GFs’ rolling.

The nature of these dependencies correlated with results from other authors [7,30,57,58,59], but differed quantitatively. It suggests common phenomena during the GFs’ compaction, while the numerical differences depended on the properties of the initial graphite, the GFs’ production method, and meso- and microstructural changes.

The differences between the *S*, *M*, *L* series both for the RD and TD at fixed densities was small, which indicated a weak effect of the GFs’ thickness on the specific electrical and thermal conductivity. This observation allowed the use of the average conductivity values. However, at high densities, the recorded value of electrical conductivity anisotropy exceeded the measurement error, which indicated structural anisotropy during the rolling process. The non-linear dependence of the thickness drop with increasing density provided indirect evidence. When the density increased from 0.70 g/cm^3^ to 1.75 g/cm^3^ the material was elongated by 5–7% and expanded by 1–2%.

### 3.2. Mesostructural Model of the Phenomenon Description of Thermal and Specific Electrical Conductivity

To describe the process of increasing thermal and specific electrical conductivity and the effects of the mesostructure on the GFs’ properties, we considered the mesostructural model [60], which was further modified and simplified for this study.

The GFs’ density was increased during the rolling process. As the rolling degree was increased, the TEG particles were changed from worm-like particle shapes to flatter ones and were aligned along the rolling plane, acquiring a preferred orientation. Figure 8a schematically represents the TEG compaction process. It was assumed that above a certain stiffness limit the worm-like particles became thinner and longer.

As the particles came closer together it resulted in the GFs’ compaction. Simultaneously, the elongated particles started to stick together, creating contact sites as shown in Figure 8b schematically for the GF density of 0.70 and 1.75 g/cm^3^. The different adherence of TEG particles with varying contact site sizes and positions can be observed, and is demonstrated by the TEM images (Figure 8c). The angle between the particles is decreased and the contact area size is increased with the density growth.

The particles are represented as cuboids for the model. The sides that touch each other are those that create contact sites. These sites are positioned at some angle *α* relative to each other along the GFs’ normal direction (ND). If the particles are equal in size, they are distributed isotopically during deformation.

This implies that the contact sites’ distribution in the specific section of the rolling plane is also isotropic (Figure 9a).

However, the graphite particles were an elongated shape. Significant stresses arose between them which oriented the particles’ long sides along the RD due to rolling. This orientation became more pronounced as the GFs’ density rose (Figure 8b,c). The contact sites formed during this process were also aligned along the RD (Figure 9b,c). The contact length along the RD was more than that along the TD. This alignment was led to increased thermal and electrical conductivity along the RD. Concurrently, the contact spots’ densities were increased as the particles’ long sides were aligned more along the RD (Figure 9c), which enhanced the thermal and electrical properties along the RD while similar values along the TD were decreased. Based on the assumption that a particle was a cuboid, its area can be represented by Equation (22) and the contact area by Equation (23):(22)$$<S_{part}>=\frac{1}{2}\cdot a\cdot l\cdot h,$$
(23)$$<S_{cont}>=a\cdot a\cdot \cos\alpha ,$$
(24)$$n_{meso}=\frac{<S_{cont}>}{<S_{part}>}=\frac{2\cdot a\cdot cos\alpha }{l\cdot h},$$
where ${<S}_{part}>$ is the average surface area of the TEG particle; a is the particle width; l is the particle length; h is the particle thickness; ${<S}_{cont}>$ is the average contact area of the particles; α is the angle between the particles; and $n_{meso}$ is the mesostructural coefficient.

Based on this, it was possible to derive the mesostructural coefficient (Equation (24)) that reflected the effects of particle densities and sizes on the foils’ properties. This coefficient allowed us to assess how thermal and specific electrical conductivity values differed at various densities, specifically between 0.70 g/cm^3^ and 1.75 g/cm^3^. Experimental and calculated (with mesostructural coefficient) data were different from each other (Table 2).

In addition, experimental data were compared to detect correlations in the properties ratios of dense foil to less dense foil: for thermal conductivity—$\lambda_{1.75}/\lambda_{0.75}=2.80$, for electrical conductivity—$\sigma_{1.75}/\sigma_{0.75}=2.71$ and for mesostructural coefficient—$n_{meso}^{1.75}/n_{meso}^{0.70}=5.22$. The differences in these ratios indicated that the mechanisms of thermal and electrical conductivity are more complex and depend on other factors.

### 3.3. The Nanostructural Factor in the GFs

#### 3.3.1. Crystallographic Structure of GF Nanocrystallites

XRD analysis of GFs was used to study the structural characteristics. For this purpose, the XRD patterns of *S*, *M*, and *L* foils with densities 0.70, 1.30, and 1.75 g/cm^3^ were taken (Figure 10) from the rolling plane. From the XRD patterns analysis, the conclusion was drawn that the all series foils have the preferred orientation of the crystal lattice basis planes of graphite along the rolling plane. It was based on the presence of the integral intensity of the Bragg reflection (00.2) and the reflection’s absence from the prismatic planes ((10.0) and (10.1)) of the hexagonal cell.

The density of GFs was increased so flat graphite mesoparticles were oriented parallel to the rolling plane during by the rolling process. The XRD analysis for all series showed the integral intensity increase from the (00.*l*) planes with the density growth. This intensity increase was associated with the more ordered orientation of graphite mesoparticles and consequently the basal plane nanocrystallites in the rolling plane, which significant changed in the XRD patterns. In this case, the S series foils at the density 1.75 g/cm^3^ was not significantly changed because the mesoparticles’ ordering was weakly expressed due to the high stress state in small-thickness GFs.

#### 3.3.2. The Misorientation Angles of GF Nanocrystallites

In our previous study [42], the misorientation angles of graphite nanocrystals around the reference normal to the plane (00.6) in foils were studied. Figure 11a shows the misorientation angle distribution depending on foils’ thicknesses in the rolling plane along the RD and TD. With the density increase in *S*, *M*, *L* series foils there was a decrease in the misorientation angles due to the nanocrystallites’ reorientation towards the preferred orientation of their basal planes due to the rolling of GFs. The color-highlighted areas depict the distribution of misorientation angles values between the RD and the TD.

At the same time, there was a trend towards a decrease in the angles, which was indicated the nanocrystallites’ reorientation in the material due to the material rolling towards the preferred orientation of their basal planes along the RD (Figure 11b).

#### 3.3.3. The Coherent Scattering Region Sizes of Nanocrystallites in GFs

It was possible to calculate the CSR sizes only along the c-direction in the graphite crystal lattice by XRD. Figure 12a presents the study results of CSRs of nanocrystallites in foils of different thicknesses and densities which were calculated by the Williamson–Hall method.

It was concluded that there was a reorientation of graphite mesoparticles during rolling by the results analysis. It was seen on the TEM images of samples *L*-0.70 (Figure 12b) and *L*-1.75 (Figure 12c). The particles were lined up in the rolling plane so more nanocrystallites entered the reflecting position, which had apparently larger size, associated with the CSRs’ size increase.

When the GFs’ thickness was changed at the density 0.70 g/cm^3^, the sizes of the nanocrystals in the CSRs remained almost unchanged across all series. This constancy was due to the GFs’ high porosity, which “prevented” significant mesoparticle reorientation. At the density 1.30 g/cm^3^ there was an increase in the size of CSRs for all series compared to the density 0.70 g/cm^3^ (Figure 12a). At the same time, there was the increase in the nanocrystallites’ size with the decrease in the GFs’ thickness (from *L* to *S*) that was associated with the graphite mesoparticles’ reorientation, and as a consequence the exit into the reflecting plane of additional nanocrystallites which had larger sizes apparently caused by the stress–strain state changes. A slightly different picture at the density 1.75 g/cm^3^ was observed. For *L* and *M* series foils, there was a tendency for nanocrystallites’ size to increase at a density of 1.3 g/cm^3^. However, for the *S* series GFs there was a sharp decrease in the CSRs’ sizes. This feature, apparently, was associated with the fact that the pressure caused by cold rolling on the mesoparticles became strong enough for a plastic deformation mechanism such that destruction and crushing of graphite nanocrystals occurred. At the same time, the graphite mesoparticles’ reorientation was difficult due to the complex stress–strain state.

#### 3.3.4. Texture Studies of the GFs

The GFs’ textural features on the D8 Discover diffractometer (Bruker, Billerica, MA, USA) with the polycapillary optics system (POLYCAP) for the formation of the small-section beam with high intensity were studied. The registration of incomplete DPFs {00.2}, {10.1}, {10.3}, {11.2} of the *L* series GFs was carried out. From this, the ODF was restored according to spherical functions (Figure 13).

According to the ODF, it was clearly seen that the axial component of the (00.1) crystallographic component was enhanced and the preferred orientation of the basal planes along the RD was intensifying. The complete DPFs {00.1}, {10.4} and {11.2} were calculated (Figure 14) from the ODF to expect the quantitative parameters of the crystallographic texture.

According to the obtained complete DPFs, it was seen that all samples have the preferred orientation of the (00.1) basal plane normal lines perpendicular to the foils’ rolling planes. The most pronounced texture was observed in the *L*-1.75 sample, which was seen from the increase in the pole density of the (00.1) maximum.

When analyzing the foils’ texture data, it was also possible to note the slight asymmetry of the texture maximum at the complete DPF {00.2} for density 1.75 g/cm^3^. With the density increase in the TD-ND-TD section of the pole figure, it has the larger integral sum of pole densities compared to RD-ND-RD (Figure 15a,b).

This effect was due to the fact that anisotropy of the GFs’ properties occurred along the RD and TD during rolling. The smaller values of this integral characteristic corresponded to the smaller nanocrystallites’ misorientation. As such, there was a more orderly orientation of the basal planes along the RD in the rolling plane which tended more strongly towards the (00.1) ideal orientation. A similar character was observed along the TD. However, the misorientation decrease was much slower compared to the RD that resulted in the anisotropy increase.

### 3.4. Anisotropy of the GFs’ Physical and Mechanical Properties in the Rolling Plane

The change in the specific electrical and thermal conductivity was slightly dependent on the thickness of the studied GFs (series *S*, *M*, and *L* at the fixed density). However, from their distribution nature it apparently followed that changes occurring inside the foils during rolling were mesostructural (change in GFs’ density and, as a consequence, porosity) and nanostructural (the nanocrystallites’ crystallographic orientation and size). As a result, high anisotropy of properties in GFs was observed.

The anisotropy study of physical and mechanical properties was conducted on the *S*, *M*, and *L* series GF samples with densities from 0.7 to 1.75 g/cm^3^. The anisotropy of mechanical properties have been previously described in [42]. The anisotropy coefficient dependence of the tensile strength of GFs with different thicknesses on their density is shown in Figure 16a.

As mentioned earlier, specific electrical resistance had high anisotropy relative to the RD and TD. The anisotropy coefficient is shown in Figure 16b, which expresses the ratio of this property’s values along the RD to the TD depending on the GFs’ density.

This coefficient was increased according to the power law by these density growths. It can be assumed that the linearity of the anisotropy coefficient with the density increase in the foils becomes more apparent. The property tended to grow more significantly along the RD due to the nanocrystallites’ preferred orientation during cold rolling. Figure 16a,b also show that the anisotropy coefficient decreased with the decline in the samples’ thickness. It was associated with the reduction in the graphite mesoparticles’ reorientation. This result also correlated with the anisotropy coefficient distribution of the misorientation angles, shown in Figure 16c. It followed from the data set that the properties’ values were influenced by the aggregate meso- and nanostructural factors.

The quantitative characteristics of the crystallographic texture were the Kearns parameters that were calculated from the incomplete DPFs data in the RD, TD, and ND, which were integral anisotropy characteristics for materials with a hexagonal crystal lattice. Texture parameter calculations are presented in Table 3.

The anisotropy by Kearns parameters was also correlated with experimental results. Crystallographic texture is a characteristic that has both meso- and nanostructural factors. However, for complicated 3D materials (as GFs undoubtedly are) it was necessary to consider the sample density, the nanocrystallites’ dimensions, and the their misorientation angles.

### 3.5. Thermal and Electrical Conductivity Given Meso- and Nanostructural Factors

The calculation of thermal and specific electrical conductivity was carried out given the quasi-monocrystal (pyrolytic graphite) properties with values along the crystallographic *a*-axis—$\lambda_{a}$ = 2000 W/(m·K) and $\sigma_{a}$ = 2500 kS/m, and along the *c*-axis—$\lambda_{b}$ = 20 W/(m·K), $\sigma_{b}$ = 10 kS/m. To calculate for porous materials, Equation (25) was modified by an experimental structural coefficient $n_{X}$ to give meso- and nanostructural transformations in the material. The experimental coefficients for thermal and electrical conductivity are presented in Equations (26) and (27), respectively.
(25)$$X_{\psi }=n_{X}\cdot (f_{c}\cdot X_{c}+(1-f_{a,b})\cdot X_{a,b})$$
(26)$$n_{\lambda }=\frac{1}{4}\cdot \cos\left(MA\right)\cdot \frac{d}{d_{Gr}},$$
(27)$$n_{\sigma }=\frac{1}{2}\cdot \frac{1}{4}\cdot \cos\left(MA\right)\cdot \frac{d}{d_{Gr}},$$
where $X_{\psi }$ is the certain property in the direction ψ; $X_{c}$, $X_{a,\;b}$ are the value of the single crystal property along the *c*- and *a*(*b*-)-axes, respectively; $f_{c}$, $f_{a,\;b}$ are the values of the Kearns parameters along the *c*- and *a*(*b*-)-axes, respectively; $n_{X}$ is the experimental structural coefficient for the property *X*; $n_{\lambda \;}$ is the experimental structural coefficient for thermal conductivity; $n_{\sigma \;}$ is the experimental structural coefficient for electrical conductivity; and $d_{Gr}$ is the X-ray density of the graphite.

However, GFs have their own characteristics that cannot be given only with these parameters. To take this into account, clarifying coefficients for thermal conductivity $n_{\lambda \;}$ (Equation (26)) and electrical conductivity $n_{\sigma \;}$ (Equation (27)) were obtained. The calculated data considering the coefficients are given in Table 4.

The calculated data were largely correlated with the experimental values that indicated the complex physical and mechanical properties’ dependence on GFs’ meso- and nanostructural factors. At the same time, the observed difference between the experimental and calculated values was apparently due to the nature of the distribution inhomogeneity of graphite mesoparticles in the GFs’ volume which was not given in this calculation, and the presence of an additional correction factor when considering the nanocrystallites’ sizes, since graphite quasi-monocrystals (their parameters were taken into account in the calculation) usually have crystallite sizes of the order of several micrometers.

## 4. Conclusions

This study investigated the physical and mechanical properties of graphite foils (GFs) of various thicknesses, produced using nitrate technology at densities ranging from 0.70 to 1.75 g/cm^3^. It was demonstrated that an increase in density results in enhanced thermal conductivity, from 109 to 326 W/(m·K), and specific electrical conductivity, from 69 to 192 kS/m. These properties are significantly influenced by the shape of the graphite mesoparticles and their arrangement during the rolling process.

The mesostructural model was proposed to explain the transfer of heat and electric current in the GFs, considering the shape, orientation, and contact areas of the particles during rolling. However, this model was weakly consistent with the experimental results. Then, it was shown that the thermal and electrical properties depend not only on mesostructural factors but also on nanostructural factors.

XRD analysis revealed that in GFs, an increase in coherent scattering regions occurred with a growth in density from 0.70 g/cm^3^ to 1.75 g/cm^3^ and a decrease in thickness from 750 μm to 200 μm. This increase is likely due to the reorientation of graphite mesoparticles during rolling, resulting in the entry of larger nanocrystallites into the reflecting plane. For foils with a thickness less 200 μm and density values more than 1.70 g/cm^3^, there was a decrease in mesoparticle reorientation and an increase in the plastic deformation mechanism of graphite, which was expressed by a sharp drop in the CSRs’ sizes.

The rocking curves showed a decrease in misorientation angles in nanocrystallites from 27 to 22 along the RD nm and from 28 to 24 along the TD nm with a growth in density from 0.70 g/cm^3^ to 1.75 g/cm^3^ during rolling. The difference in values in the RD and TD was due to anisotropy, which affected the distinction of the electrical and mechanical properties of rolled GFs in the main directions.

The preferred crystallographic orientation of nanocrystallites arising from rolling was studied by the X-ray method. It was shown that the axial rolling texture was formed for all samples with the (00.1) maximum increase as the foils’ density was increased. According to the obtained data, the Kearns parameters were determined on which were calculated the anisotropy coefficient A, for the electrical and thermal conductivity of L series foils by different densities: A_0.70_ = 1.05, A_1.30_ = 1.10, and A_1.75_ = 1.16. These calculated values were consistent with experimental measurements of anisotropy coefficients for specific electrical conductivity (from 1.00 to 1.16) and mechanical properties (from 0.98 to 1.13).

Structural coefficients were identified, allowing the inclusion of meso- and nanostructural factors in the calculations of thermal and electrical properties. The calculated values largely correlated with the experimental results for thermal and specific electrical conductivity. The maximum discrepancy between the experimental data for thermal conductivity was 6.0%, and for electrical conductivity it was 9.0% in the rolling direction and 16.6% in the transverse direction. To improve the convergence of the results, the mechanisms of changes in the meso- and nanostructure of graphite foils that ensure the formation of the final properties will be examined and applied to the model in more detail in future studies.

Subsequent research on graphite foils will focus on further investigating the mechanisms behind the anisotropy of their properties. This includes determining the impact of various thermo-mechanical treatments on the final properties, and identifying the mechanisms for producing materials with specific characteristics, particularly electrical and thermal properties. Once the mechanisms for achieving graphite foils with the desired properties are sufficiently understood, future studies will aim to develop predictive models of their behavior during rolling and other thermo-mechanical processes.

## Figures and Tables

**Figure 1 nanomaterials-14-01162-f001:**
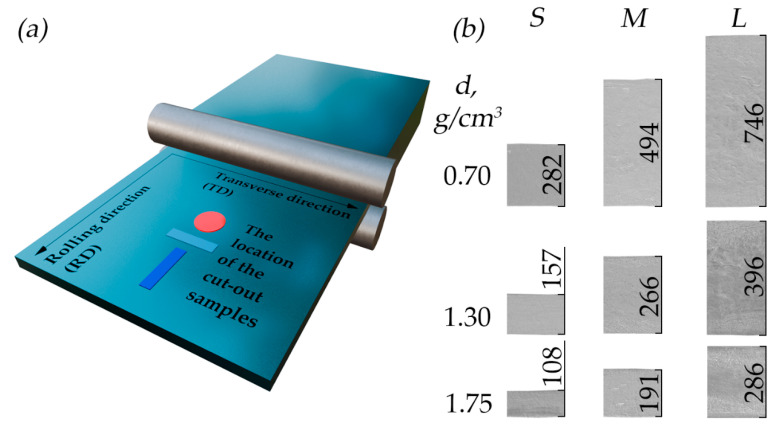
(**a**) Schematic image of the GF rolling process with an indication of the sample cutouts; (**b**) images of front cross-section of foils across rolling at densities of 0.70, 1.30, and 1.75 g/cm^3^ using SEM.

**Figure 2 nanomaterials-14-01162-f002:**
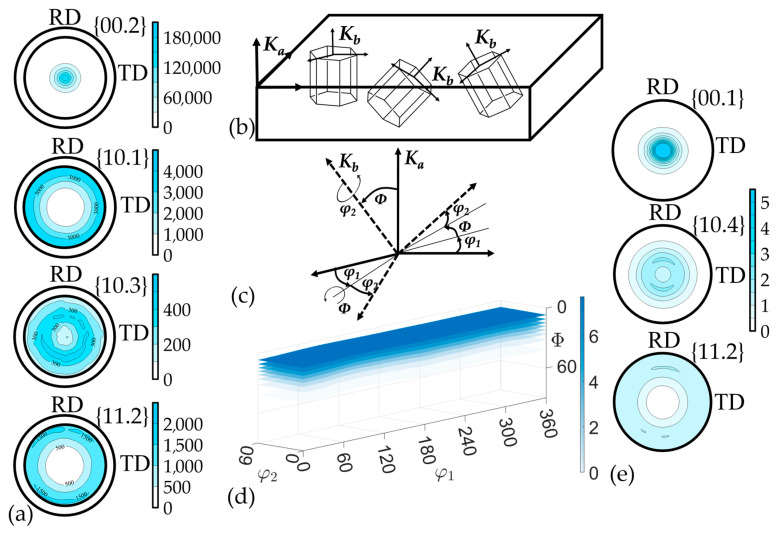
Determination of crystallographic texture: incomplete DPFs (**a**), relation between the $K_{a}$ sample coordinate system and the $K_{b}$ grains’ crystallographic orientation system (**b**), Euler angles (**c**), ODF (**d**), restored complete DPFs (**e**).

**Figure 3 nanomaterials-14-01162-f003:**
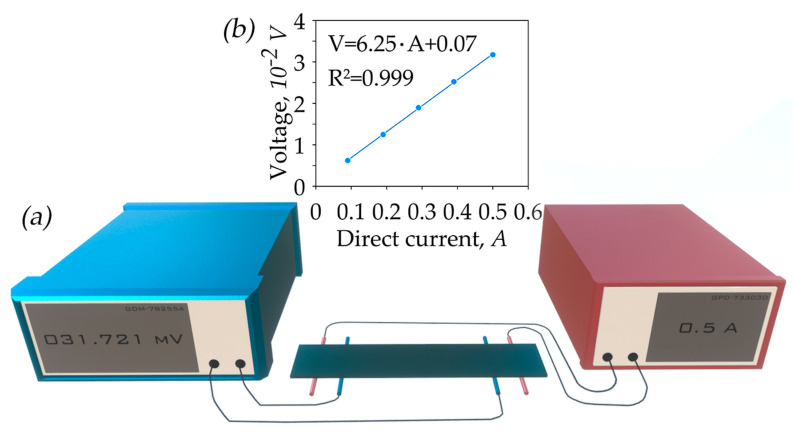
Diagram of the test bench (**a**); current–voltage characteristic (**b**).

**Figure 4 nanomaterials-14-01162-f004:**
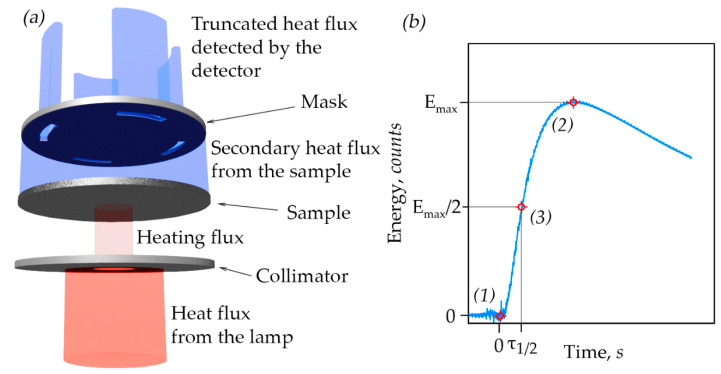
(**a**) The principle of operation of LFA In-plane; (**b**) the type of experimental curve.

**Figure 5 nanomaterials-14-01162-f005:**
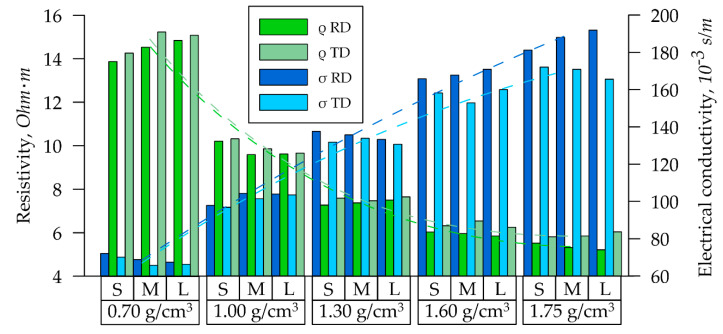
Dependence of the electrical characteristics of GF samples by their series.

**Figure 6 nanomaterials-14-01162-f006:**
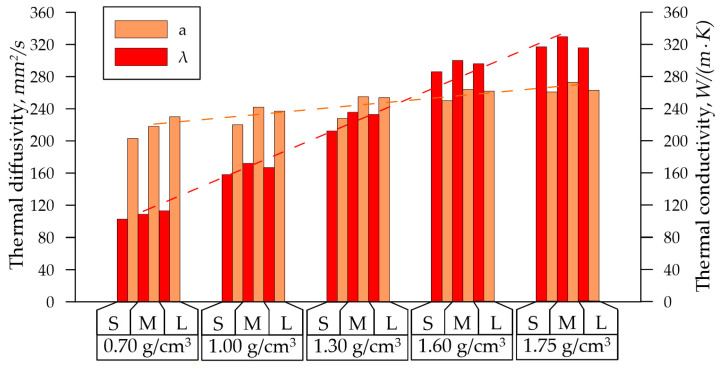
Dependence of the GFs’ thermal properties on density.

**Figure 7 nanomaterials-14-01162-f007:**
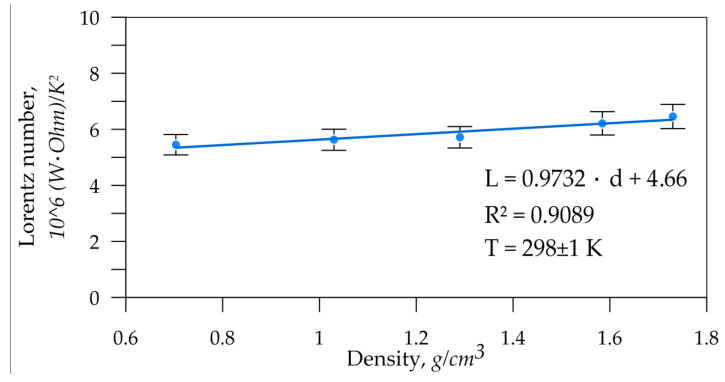
Dependence of the Lorentz number (*L*) on the GF density.

**Figure 8 nanomaterials-14-01162-f008:**
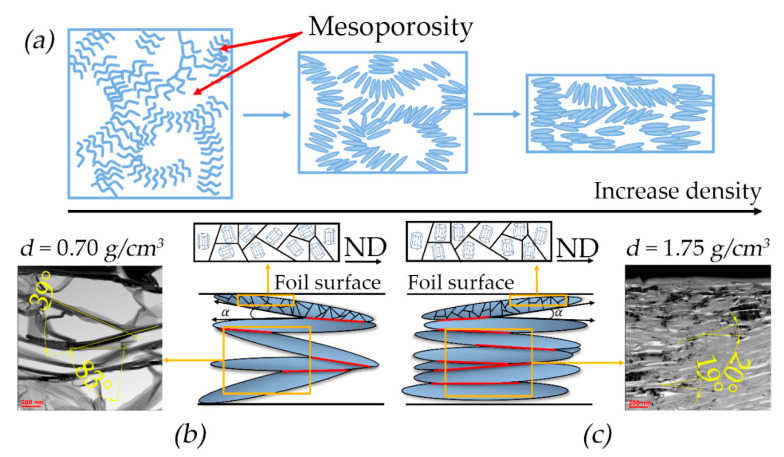
The mesostructural model: (**a**) laying of graphite flakes during rolling; laying of graphite flakes at the densities 0.70 g/cm^3^ (**b**) and 1.75 g/cm^3^ (**c**).

**Figure 9 nanomaterials-14-01162-f009:**
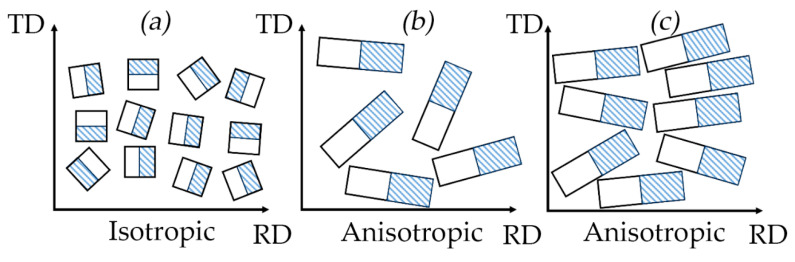
Schematic representation of sections in the rolling plane with the image of particles and contact sites (bar line) of isotropic porous material (**a**) and anisotropic GFs with densities of 0.70 g/cm^3^ (**b**) and 1.75 g/cm^3^ (**c**).

**Figure 10 nanomaterials-14-01162-f010:**
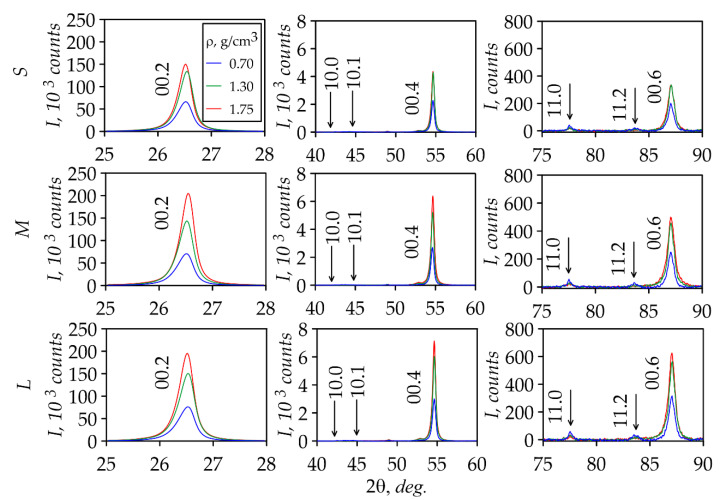
XRD patterns of foils of the *S*, *M*, and *L* series.

**Figure 11 nanomaterials-14-01162-f011:**
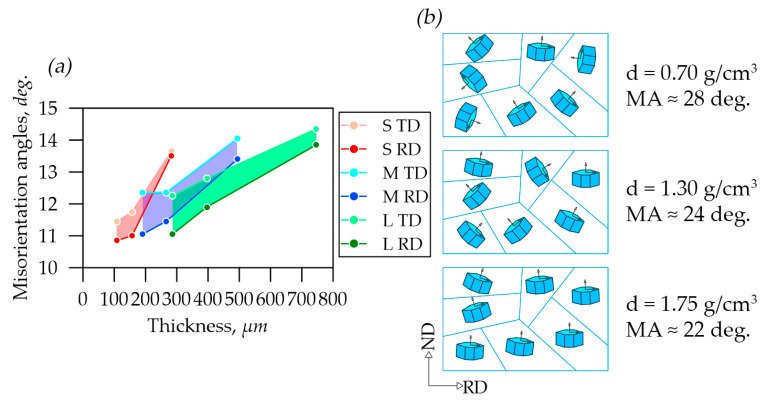
Misorientation angle distribution in graphite foils (**a**) and schematic images (**b**) of the nanocrystallites’ misorientation along the rolling plane in a separate graphite flake.

**Figure 12 nanomaterials-14-01162-f012:**
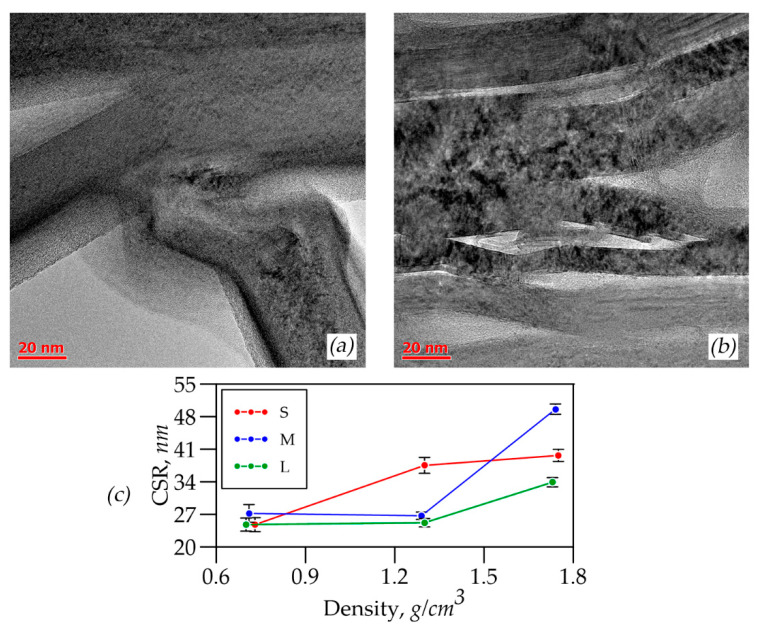
TEM images of GFs with densities of 0.70 (**a**) and 1.75 (**b**), CSR distribution depending on the densities of S, M, and L foils (**c**).

**Figure 13 nanomaterials-14-01162-f013:**
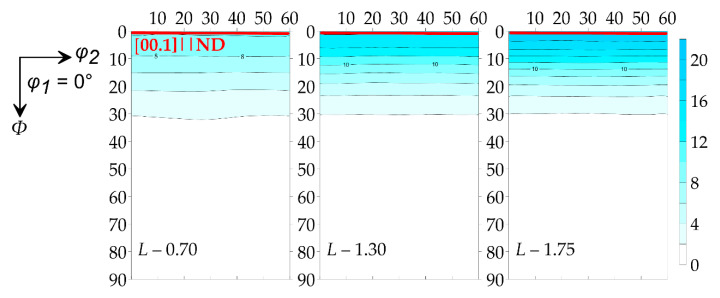
ODF of samples of foils of different densities.

**Figure 14 nanomaterials-14-01162-f014:**
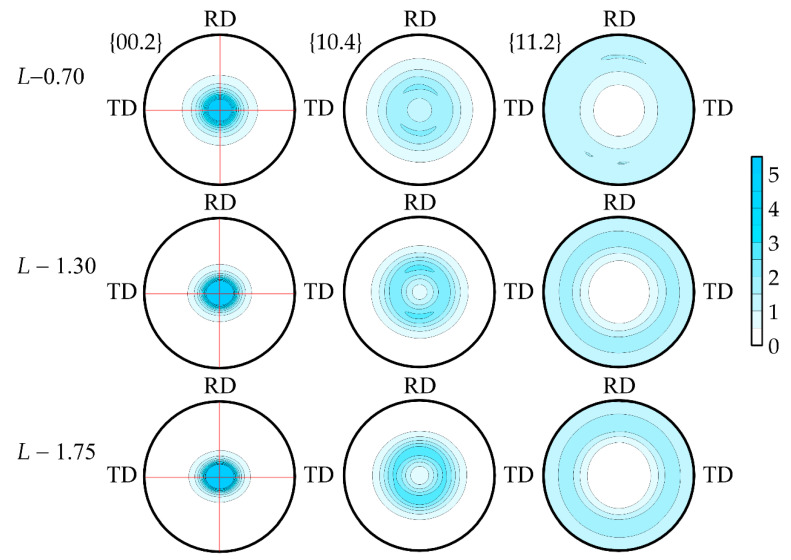
Full DPFs of the test samples *L*-0.7, *L*-1.3 and *L*-1.75.

**Figure 15 nanomaterials-14-01162-f015:**
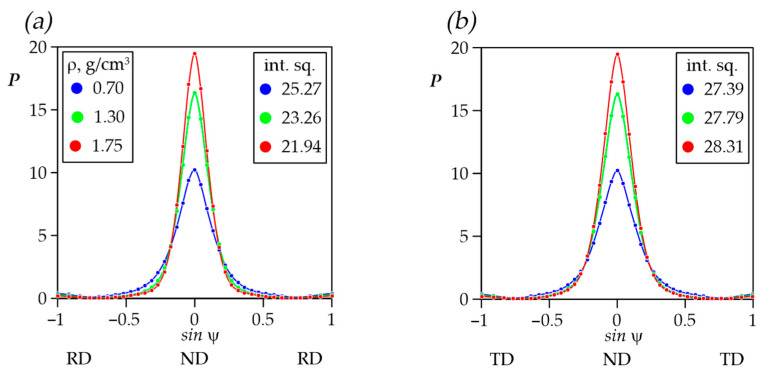
Distribution of pole density in sections of complete DPF {00.2} RD-ND-RD (**a**) and TD-ND-TD (**b**).

**Figure 16 nanomaterials-14-01162-f016:**
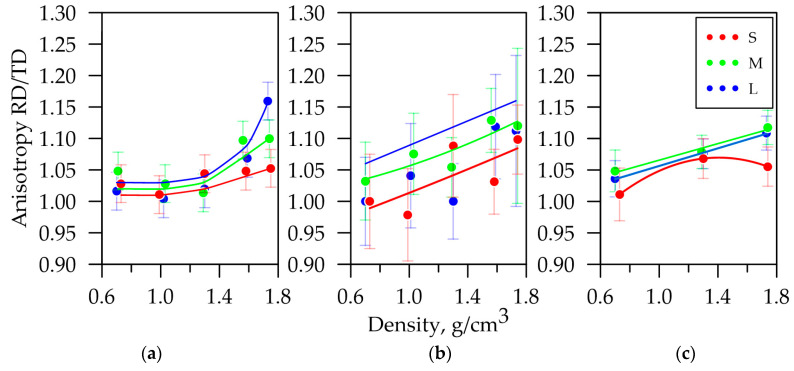
The anisotropy coefficient dependence of the electrical conductivity (**a**), the tensile strength (**b**), and misorientation angles (**c**) on the GFs’ density.

**Table 1 nanomaterials-14-01162-t001:** Parameters of the tested samples.

Series *S*		Series *M*		Series *L*	
h, μm	d, g/cm^3^	h, μm	d, g/cm^3^	h, μm	d, g/cm^3^
282 ± 5	0.73 ± 0.01	494 ± 7	0.71 ± 0.03	746 ± 8	0.70 ± 0.01
207 ± 3	0.99 ± 0.03	342 ± 3	1.03 ± 0.07	510 ± 5	1.02 ± 0.01
157 ± 5	1.30 ± 0.03	266 ± 4	1.29 ± 0.11	396 ± 5	1.30 ± 0.01
127 ± 5	1.58 ± 0.04	211 ± 4	1.56 ± 0.02	320 ± 4	1.59 ± 0.01
108 ± 3	1.75 ± 0.04	191 ± 6	1.74 ± 0.03	286 ± 8	1.73 ± 0.02

**Table 2 nanomaterials-14-01162-t002:** Experimental and calculated values of thermal and specific electrical conductivity by only the mesostructural factor.

Density, g/cm^3^	α, °	$n_{meso}$	$\lambda_{meso}$, W/(m·K)	$\lambda_{exp}$, W/(m·K)	$\sigma_{meso}$, kS/m	$\sigma_{exp}$, kS/m
0.70	39	0.009	18	113	23	66
1.75	19	0.047	94	316	118	179

**Table 3 nanomaterials-14-01162-t003:** Kearns integral texture parameters.

Sample	Density, g/cm^3^	Thickness, µm	Integral Texture Parameters			$\mathrm{Anisotropy}\;\mathrm{RD-TD}\;\frac{f_{TD}}{f_{RD}},$ %
			*f* _ND_	*f* _RD_	*f* _TD_	
*L*-0.7	0.70	746	0.715	0.139	0.146	1.05
*L*-1.3	1.30	396	0.798	0.096	0.107	1.11
*L*-1.75	1.73	286	0.830	0.078	0.093	1.16

**Table 4 nanomaterials-14-01162-t004:** Experimental and calculated thermal and electrical conductivity.

Samples	λ_calc_, W/(m·K)		<λ_calc_>, W/(m·K)	<λ_exp_>, W/(m·K)	σ_calc_, kS/m		σ_exp_, kS/m	
	RD	TD			RD	TD	RD	TD
*L*-0.70	118	116	117	113	73	72	67	66
*L*-1.30	237	231	234	220	148	144	133	131
*L*-1.75	330	319	325	316	206	199	192	166

## Data Availability

The data presented in this study are available on request from the corresponding author.

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
