# Peer review of "Anisotropy of Electrical and Thermal Conductivity in High-Density Graphite Foils"

_nanomaterials, 2024, doi:10.3390/nano14131162_

Round 1
Reviewer 1 Report
Comments and Suggestions for Authors
The manuscript titled “Anisotropy of electrical and thermal conductivity of high-density graphite foils” by Shulyak, V.A.; et al. is a scientific work where the authors assessed the electrical, thermal and mechanical properties of graphite foils with different thicknesses and how their architecture morphology and structure can affect this parameters. For it, many complementary techniques were devoted. The most relevant outcomes found by the authors could open new gates in the design of the next-generation of smart coated surfaces with optimized electro-thermal performance which could be of interest for many Industrial applications. The manuscript is generally well-written and this is a topic of growing interest.
However, it exists some points that need to be addressed (please, see them below detailed point-by-point) to improve the scientifc quality of the submitted manuscript paper before this article will be consider for its publication in Nanomaterials.
1) ABSTRACT. The authors should erase the gaps between the different paragraphs.
2) KEYWORDS. I am not sure but the maximum number allowed of keywords could be exceeded (10). The authors should contact with the Assistant Editor to discuss about this point.
3) INTRODUCTION. “Flexible graphite foils (GFs) are a derivate of natural graphite (…) expanding the already considerable range of industrial applications for graphite materials” (page 2). The authors should provide quantitative data details about the worldwide burdens of graphite foil production and the associated economic impact on society. This will significantly aid the potential readers to better understand the significance of this devoted research.
4) “Carbon materials are headliners in thermal conductivity (…) due to its relative low cost, ease of manufacturing, and high production volume” (page 2). Even if I agree with this statement provided by the authors it should be also discussed the appealing magnetic properties of these materials under certain conditions at high temperatures [1] which could be also exploited for quantum technologies [2].
[1] Zhou, S.; et al. High-Temperature Quantum Hall Effect in Graphite-Gated Graphene Heterostructure Devices with High Carrier Mobility. Nanomaterials 2022, 12, 3777. https://doi.org/10.3390/nano12213777
[2] Winkler, R.; et al. A Review of the Current State of Magnetic Force Microscopy to Unravel the Magnetic Properties of Nanomaterials Applied in Biological Systems and Future Directions for Quantum Technologies. Nanomaterials 2023, 13, 2585. https://doi.org/10.3390/nano13182585
5) SAMPLE PREPARATION & TEST METHODS. 2.1. Production of flexible graphite foils. “The process of obtaining graphite samples (…) treated with fuming nitric acid in a weight ratio of 1:0.0 to obtain a GIC (…) foaming the exhaust gas at a temperature of 900 ºC to produce TEG” (page 2). What were the incubation times for both chemical reactions? Did the authors follow any strategy to preserve the graphite foils once produced prior the subsequent characterization steps? In case affirmative, it should be detailed in this subsection.
6) Then, the software used to process the raw data by the different techniques should be detailed in each subsection.
7) RESULTS & DISCUSSIONS. 3.1.1. Electrical characteristics of GFs. Figure 5 (page 9). The standard deviations (SD) bars should be added for each examined condition. Same comment for the Figure 6 (page 10).
8) Figure 7 (page 11). In the gathered equation “L = 0.9732 . d + 4,66” the last comma should be replaced by a point.
9) 3.1.3. The interrelation between thermal and specific electrical conductivity (page 11). Did the authors observe any resistivity effects during the data acquisition of these measurements? A brief statement should be furnished.
10) 3.2. Mesostructural model of the phenomenon description of thermal and specific electrical conductivity (page 12-14). Did the authors observed any local anisotropy effects about the thermal and electrical conductivity properties measured in the graphite flakes? Some further explanation should be provided in this point.
11) “Figure 8-a (…) TEG compaction process (…) stiffness limit the worm-like particles became thinner and longer. The particles came closer together that it was resulted to the GFs compaction” (page 12). Did the authors observe a gel-phase transition during the compaction process? In case affirmative, this could negatively affect to the data interpretation? Some discussion should be appended in this regard.
12) CONCLUSION. This section perfectly remarks the most relevant outcomes found by the authors in this work. The authors should add a brief statement to discuss about the future line actions to pursue this research and the open perspectives.
Comments on the Quality of English Language
The manuscript is generally well-written albeit it may be desirable if the authors could recheck it in order to polish those final details susceptible to be improved.
Reviewer 2 Report
Comments and Suggestions for Authors
Recommendation: Major revision’
Comments:
- The abstract was too lengthy without describing the necessity of the present investigation.
- Instead of explaining all the outcomes, it is recommended to focus only on the most significant results in the abstract.
- Introduction: The background information/necessity/scope of the present examination should be discussed properly.
- What was the specific application of the present GF study?
- The use of symbols/abbreviated terms should be regularized properly.
- End of the introduction: The information regarding the research gap and work novelty was not presented.
- Avoid using lumped references. It is recommended to use a maximum of three.
- Table 1: On what basis these dimensions have been selected? Need a justification/reference.
- An elaborated discussion about the present work limitation and future scope is recommended as a separate section.
Comments on the Quality of English Language
Minor revision is needed
Reviewer 3 Report
Comments and Suggestions for Authors
The paper is devoted for investigations of anisotropy of electrical and thermal conductivity of high-density graphite foils. The topic is generally interesting, however the paper contain unexplained places (below) and need major revisions.
Fig. 5 is not clear. Why in the same graph should be presented two reciprocal quantities: resistivity and electrical conductivity?
Please explain why for electrical properties analysis were used equations (15-18)?
Please explain the thermal diffusivity and thermal conductivity dependence on the density presented in Fig. 6.
Please explain results presented in Fig. 7.
English need minor revisions.
Conclusions should be rewritten in more informative way.
Comments on the Quality of English Language
Minor English revisions are needed.
Round 2
Reviewer 2 Report
Comments and Suggestions for Authors
Accept in present form
Reviewer 3 Report
Comments and Suggestions for Authors
Authors make proper corrections according to reviewer remarks and I suggest to publish the paper as it is.